# Condition Evaluation of an Existing T-Beam Bridge Based on Neutral Axis Variation Monitored with Ultrasonic Coda Waves in a Network of Sensors

**DOI:** 10.3390/s20143895

**Published:** 2020-07-13

**Authors:** Hanyu Zhan, Hanwan Jiang, Jinquan Zhang, Ruinian Jiang

**Affiliations:** 1Klipsch School of Electrical and Computer Engineering, New Mexico State University, Las Cruces, NM 88003, USA; hzhan@nmsu.edu; 2Department of Civil Engineering, University of Wisconsin-Platteville, Platteville, WI 53818, USA; 3Research Institute of Highway, Ministry of Transport, Beijing 100088, China; jq.zhang@rioh.cn; 4Department of Engineering Technology & Surveying Engineering, New Mexico State University, Las Cruces, NM 88001, USA; rjiang@nmsu.edu

**Keywords:** ultrasonic coda wave, data acquisition, signal processing, sensor technology, neutral axis locating, nondestructive testing, structural health monitoring

## Abstract

Neutral axis passing through the stiffness centroid of a structure is correlated with structural health conditions. Traditional techniques rely on gauge arrays to observe strains at their installation positions, and then locate a neutral axis through the intercept of the strain diagram. However, these localization results will be severely deviated if any damages exist among gauges or inside structures. In this paper, a novel technique is proposed to locate the neutral axis by measuring and analyzing ultrasonic coda waves in a network of transducers. Because of multiple trajectories, coda waves are sensitive to minor changes in a large volume of media that are not limited to direct paths between sensors. This technique is not only capable of locating a neutral axis with great efficiency and accuracy, but can also indicate global structural health and inner damages. The applicability of the technique is demonstrated by monitoring a 30 m concrete T-beam subjected to four-point loading tests. With an array of transducers placed at the surface, the neutral axes in the large region are located. The localization results also show clear trends that the global neutral axis moves up as the loads increase, which indicates the beam contains certain degrees of inner damage.

## 1. Introduction

The neutral axis refers to the curve that passes through the stiffness centroid of a structure and it does not experience any deformation within the cross-section under loading [1]. Since this centroid is only dependent on the material and geometrical properties, theoretically, the neutral axis location is constant. However, concrete structures are continually subject to mechanical, chemical, and thermal effects, which impose some types of damages on them. These damages, which result in material loss and stiffness reduction, will relocate the neutral axis. The relocation will increase stresses in the tension zone, and further worsen damages and shift neutral axis until a new balanced state is reached. Any change in the locations indicates the stiffness variations, which implies the structure has suffered from damages or deterioration. Hence, the neutral axis is a potentially powerful tool for structural health evaluation and minute damage detection if its location can be determined properly [2].

Current techniques rely on structural analysis and strain gauge measurements to locate the neutral axis. Concrete is a multi-composite material made of cements, sand grains, aggregate stones, steel bars, and porosities. The property of each composite under variable environmental effects, together with geometrical properties, construction tolerances, and damages, contribute different uncertainties in the localization [3,4]. Thus, structural analysis requires a large number of tests to estimate uncertainty statistics in composite and construction as a priori knowledge. Additionally, the analysis results in general only provide approximate ranges of average axis locations in healthy structures. On the other hand, the general idea of gauge measurements is to install gauge arrays along the vertical direction of structural cross-section to collect the strain profiles at positions where they are exactly placed. Then the neutral axis is located at the height corresponding to the zero deformation through the intercept of the strain diagram [5,6,7]. However, gauge measurements do not carry any information about the areas among their installation positions. Consequently, this technique has limited detection range closed to gauge arrays, and any nonlinear variations/damages appearing among gauges will cause the inaccurate localization results even if plenty of gauges are placed closely. Furthermore, gauges may suffer irreversible deformations and damages when cracking or significant changes take place nearby. Another major limitation of both analysis and gauge techniques is their inapplicability to inner damage detection. This issue is largely circumvented in non-destructive testing and evaluation (NDT&E) of in-service structures since sensor embedment for them is generally unavailable. Due to the brittle nature, structural failure is usually initiated by micro-cracking inside concrete at an early stage, but the neutral axis located by gauges mounted on the surface cannot give any indicators of inner damages. Over the past several decades, few techniques have been proposed to solve these problems.

Ultrasonic waves constitute one of the primary tools in concrete structural NDT&E applications [8,9]. Measuring velocity, amplitude, and nonlinearity of direct waves show promise for detecting thickness, delamination, and damages [10]. However, the high heterogeneities contained in multi-composite concrete cause strong multiple scattering behaviors during the propagation of ultrasonic waves. As a result, the energy of direct waves is significantly attenuated and mostly transformed into long-lasting waveforms arriving later in the record, which form the coda waves [11]. These features are well-known to severely affect the performance of traditional direct wave-based techniques. Fortunately, previous studies in the fields of optics, acoustic, and geophysics demonstrate that noise-like coda waves contain detailed information about internal structures, and they are very sensitive to changes occurring in the large volume of a medium [12,13,14,15,16,17,18,19,20,21,22,23,24,25,26,27]. This sensitivity is attributed to the fact that coda waves propagate following complex trajectories not limited to the direct path between two sensors, and they may traverse the regions of interest repeatedly. Thanks to these properties, coda waves have been successfully used for several NDT&E applications, mostly on laboratory scales. Depending on practical applications, these coda wave-based techniques can generally be separated into two categories: evaluation of global structural health [16,17,18,19,20] and characterization of cracks [21,22,23,24,25,26]. The first category focuses on global structure monitoring through the use of coda wave interferometry (CWI) [27] or diffusion equation [11] to quantify coda signal changes. Then, with the help of a sensor network and further analysis, such as Mahalanobis distance [18] and diffusivity [20], the location, type, and magnitude of changes can be estimated up to a certain extent. The other category aims at identifying the locations, depths, and numbers of cracks with a high degree of accuracy. Its general concept is to first calculate coda waveform variations. An inversion procedure is then adopted to calculate the density of changes at each localized position through inverse algorithms, such as maximum likelihood [23], sensitivity kernel [24,25,26], and interpolation [21]. For additional details on the techniques, we refer the reader to the literature reviews published in References [20,25].

This paper proposes a novel technique to locate the neutral axis in concrete using coda wave measurements in a network of transducers. When applied to a 30-m T-beam subject to four-point bending tests, the neutral axis in a large region are located within seconds using limited numbers of fixed transducers placed at the surface. The localization results indicate global structural health and locate multiple micro-cracks up to an applicable extent. Compared to current techniques, the main advantages of this novel one includes: 1) coda wave reflects changes occurring in a large volume, and, thus, their neutral axis locating results can indicate both surface and inner damages of concrete. 2) The technique is straightforward and efficient so that the neutral axis in a large structure can be located within seconds by using limited numbers of fixed transducers. 3) Ultrasonic transducers will not be damaged by significant changing states nearby, and they are reusable by simply gluing on surface without drilling requirements.

In this paper, Section 2 describes the methodology and Section 3 introduces the materials and equipment. The measurements and data processing are illustrated in Section 4. We present and discuss the experimental and analysis results in Section 5. Then the paper is concluded in Section 6.

## 2. Methodology

### 2.1. Principle

Strain gauge benefits from its electrical resistance dependence on geometry of concrete to infer stress changes. A changing stress state in concrete will slightly stretch or compress the internal electrical conductor of gauges, and further increases or decreases the electrical resistance end-to-end. Since concrete elasticity is relatively stable at low stress levels below serviceability, the stress changes can be estimated from the electrical resistance measurements. However, the internal conductors may encounter irreversible deformation beyond their elasticity limits if cracking or significant medium changes appear nearby.

Neutral axis can be located with a minimum of two parallel gauges in a cross-section. Figure 1 illustrates two strain measurements with one at the top *a_t_* and one at the bottom *a_b_* with distance *h* between them. Assuming a linear strain distribution, the neutral axis height *y* is the intercept of the strain diagram [6].
(1)$$y=\frac{a_{b}h}{a_{b}-a_{t}}+b$$

For three or more parallel gauge measurements, the location of the neutral axis can be determined by simply averaging the positioning results from all available gauge pairs, or by finding the best fit lines based on least-square and maximum-likelihood algorithms [28].

Since gauges can only measure the strain changes at positions where they are exactly placed. Thus, their localization results of neutral axis will be inaccurate if any damages or non-linear variations appear among the gauges. In addition, the results also cannot give any indication of inner damages. On the contrary, coda waves propagate following complex multiple scattering trajectories that are not limited to the direct path between two sensors, and they may traverse the regions of interest repeatedly (see Figure 2). The unperturbed coda wave *E_u_*(*t*, *r*) observed at a given position *r* is a sum of the partial waves *A_X_*(*r*, *t*) from all the trajectories *X* in a medium.
(2)$$E_{u}\left(t,r\right)=\sum^{\text{​}}A_{X}\left(t,r\right)$$

When medium changes over time *t*, the dominant effect is a variation in the arrival time *∆t_T_* (or speed ∆*v_T_*) of each partial wave. The perturbed coda wave *E_p_*(*t, r*) can be expressed by the equation below.
(3)$$E_{p}\left(t,r\right)=\sum^{\text{​}}A_{X}\left(t-∆t_{T},r\right)$$

In a summary, coda wave has traveled a large volume for a long time so that small changes anywhere in media result in a notable absolute time lag, which makes it ideal for NDT&E applications. Figure 3 shows the great sensitivity of coda waves following a medium change as an example. The blue curve represents the unperturbed wave, and the red curve represents the perturbed wave under a small load. Coda waves at the late part of the signal (window 2) show obvious phase changes, whereas the direct waveforms in the single scattering regime (window 1) are nearly invariable.

### 2.2. Coda Wave Interferometry

Code wave interferometry (CWI), which is a technique initially developed from seismology [29], was recently extended to compare ultrasonic waveforms in concrete. It is based on computing the degree of similarity of the unperturbed and perturbed coda waves [27].
(4)$$CC=\frac{\int_{T-T_{w}}^{T+T_{w}}E_{p}\left[t\left(1+\varepsilon \right)\right]E_{u}\left(t\right)dt^{\prime }}{\sqrt{\int_{T-T_{w}}^{T+T_{w}}{\left\{E_{p}\left[t\left(1+\varepsilon \right)\right]\right\}}^{2}dt\int_{0}^{T}{\left[E_{u}\left(t\right)\right]}^{2}dt^{\prime }}}$$
where
(5)$$\varepsilon =\frac{∆t}{t}=-\frac{∆v}{v}$$

In this case, the cross-correlation *CC* is a function of the dilation rate *ε*. As shown in Figure 3, the results of the velocity change ∆*v*/*v* in coda waves can be considered as dilation in time (i.e., time change ∆t/t). The *ε* value is determined by maximizing the *CC*, i.e., making the perturbed wave in a chosen time window best resemble the associated unperturbed wave. *T* and *T_w_* are the center time and half width of the time window, respectively.

Interpreting the *ε* and *CC* results are able to distinguish different types of changes in concrete. If stress changes exist by selecting appropriate *ε* values, the dilation effects between waveforms are removed and *CC* values should be large. In this case, the compression or dilation of the perturbed signal with negative or positive *ε* values indicates a decrease or increase of propagation velocity. However, damages worked as extra scatters will cause waveform distortions. Consequently, *CC* should be kept at low values no matter what *ε* value is selected.

Then to locate the neutral axis, lines connecting all available receiver pairs are used to determine the coordinates corresponding to the stress invariant (i.e., *ε* = 0) through linear intercepts of *ε* diagrams. This procedure looks similar to the strain intercepts illustrated by Equation (1). However, *ε* values calculated from coda wave measurements carry information about changes and damages occurring in a large volume of media. Thus, this method can locate the axis at positions without placing transducer arrays, and the localization results are highly correlated with both surface and inner damages.

## 3. Materials and Equipment

### 3.1. Concrete T-Beam

The concrete T-beam weighing about 65 tons was removed from a bridge after being in service for more than 15 years. It was made of typical materials of No. 40 concrete (40 MPa specified compressive strength), reinforcements, and pre-stressed strands. The lifting was performed by two overhead cranes on the truck, and then the T-beam was transported to a National Bridge Structure Safety Engineering Lab, Beijing and installed on its supports.

The T-beam is 30 m in length and 1.6 m in height. Its widths at the top and bottom surfaces are 1.7 and 0.4 m, respectively. Figure 4 shows the detailed dimensions. During the service period, the T-beam has experienced a large amount of mechanical loads, corrosions, and environmental effects, and it can well represent plenty of in-service large concrete structures in the real world.

### 3.2. Ultrasonic Transducers

The ultrasonic transducer RS-2A designed by Softland Times Scientific & Technology Co. Ltd. has broadband working frequency of 60∼400 kHz. It is 15 mm in thickness and 18.8 mm in diameter. As shown in Figure 5, a total of 14 RS-2A transducers were glued on the east web near the center line using the coupling medium of high vacuum silicone grease. During the experiment, this region was checked carefully and no cracks appeared, whereas the opposite west face contained several pre-existing cracks. The sizes of these cracks were measured by optical observations with a microscope and a simple rule, and their widths were on the order of a millimeter. Experimental results described in Section 5 will show that these micro-cracks are detected and located even though no transducers were placed at the west face. In this case, six transducers S1∼S6 working as sources were placed horizontally at 90 cm height, and the distance between two adjacent sources is 60 cm. Eight transducers served as receivers were positioned along two parallel lines: R1∼R4 and R5∼R8 were placed in grids at 60 cm and 113 cm heights, respectively. We note that there is no need for perfect regular spacing, and operators can place receivers at their own convenience.

In addition, 10 strain gauges G1∼G10 were installed along two vertical lines at the heights of 55, 75, 95, 115, and 130 cm. Additionally, the concrete elasticity modulus are estimated to be 32,500 MPa.

### 3.3. Excitation and Measurement Equipment

A high power ultrasonic system RITEC SNAP 5000 was connected to each source to successively emit the identical excitation impulse. The response signals detected by each receiver were first amplified by a Smart AE amplifier, and then collected with a customer-made NI PXI-1000 Data Analyzer that can measure up to eight channels simultaneously. Each source-receiver pair has an independent time trigger and counter to accurately record the excitation and observation time. Furthermore, strain gauge observations were performed with a DH3821 multi-channel static strain test and analysis system.

## 4. Acoustical Measurements

### 4.1. Measurement Procedure

Figure 6 shows the four-point bending test conducted on the T-beam. Two hydraulic jacks were placed symmetrically close to the mid-span, and the distance between them is 450 cm. The loads of each jack were increased from 0 to 50 kN with a fixed step of 10 kN.

The excitation ultrasonic impulse was selected at 150 kHz with 300 Vpp excitation voltages. Higher frequencies corresponding to shorter waves are more sensitive to weak changes in media but will enhance wave energy attenuations due to scattering and dissipation effects [11]. The frequency chosen at 150 kHz is a trade-off result between the detection capability and sensitivity based on several trials, and this value is consistent with the typical frequencies used to generate coda waves in concrete [11,15]. Furthermore, using single-frequency impulses, as opposed to transient bursts, can simplify the data processing and increase the transmission range. The sampling frequency was set at 2.5 MHz frequency that conforms to the Nyquist sampling theorem.

At each loading step, successive excitations-acquisitions of wave signals were made for all available source-receiver pairs. Specifically, for each measurement, only one source was activated to generate the impulse every 6 milliseconds, and all the receivers continuously measured the response signals. In order to increase signal-to-noise ratios (SNR), the excitation was reproduced 50 times and the measurements were recorded consecutively within 0.3 s. Then, this source was turned off and the next source was switched on to emit the identical impulses 50 times, and the measurements were repeated for all the receivers. Thus, each loading step includes 48 measurements corresponding to 6 × 8 source-receiver pairs and each measurement contained 50 full coda waveforms. Since the signals are observed in an extremely short time, the environmental conditions can be treated as constants. As a result, the waveforms within each cycle have an extremely high degree of similarity, and their correlation values generally are greater than 0.998.

### 4.2. Data Processing

The data processing is straightforward without any requirement of frequency filtering and time-window selection. For each source-receiver pair, 50 cycles of measurements within 0.3 s at each loading step are averaged. Then, the average full waveform at 0 kN load is used as the reference (i.e., unperturbed waveform) to quantify the waveform variations at another loading step (i.e., perturbed waveform) through Equation (4). In this case, directly using the full waveforms instead of a specific part not only simplifies the data processing but also proves the generality of the proposed technique.

Figure 7 shows the average waveforms corresponding to the S1-R1 and S1-R8 pair measurements at 0 kN and 20 kN loads. Similar to Figure 3, a part of the waveforms is enlarged to denote the details of their variations caused by the load. For the S1-R1 pair, the time compression is found in the waveform at 20 kN relative to its reference waveform, which implies compression forces play a major role in this region. In contrast, the clear time dilation is observed in the S1-R8 waveforms indicating tension effects.

## 5. Results and Discussions

In Section 5.1, the stress changing status from structural analysis as well as strain gauge and coda wave measurements are evaluated. In Section 5.2, the neutral axis locations determined by coda wave observations are presented to evaluate the global structural health. The associated results from structural analysis and gauge measurements are also presented for comparison. In Section 5.3, the coda wave-based localization results are further analyzed to indicate the crack positions.

### 5.1. Evaluation of Stress Changes

Figure 8 shows the structural analysis values of the stress changes at the receivers R1∼R8 positions. A finite element model was established to obtain the results where a linear elastic stress-strain relationship was adopted for the analysis. The modeling results also agree well with the results calculated with the classic stress-bending moment equation. For R1∼R4 placed in the upper portion of the T-beam, their locations experience the changing compressive stresses that increase linearly with the loads. On the other hand, the changing tensile stresses, appearing at the R5∼R8 positions of the lower portion, also linearly increase as the loads increase.

The stress changing values measured with the G1∼G10 strain gauges are illustrated in Figure 9, which are in basic agreement with the analysis results. For the G1, G2, G9, and G10 gauges placed in the lower portion, their stress changes decrease almost linearly as the loads increase. At the same time, the stress changes at the G4∼G7 positions increase. The stress changes at the G3 and G8 positions are close to 0, which implies the neutral axis is located near them.

Figure 10 presents the *ε* calculation values corresponding to the source S1 excitation. They show the same trend with the analysis results depicted in Figure 8, i.e., the *ε* values of R1∼R4 and R5∼R8 increase and decrease approximately linearly with the loads. These results prove that *ε* values (i.e., coda waveforms) can well recognize compression and tension stress changes in concrete. The *ε* results from the source S2∼S6 excitations are also in accord with this trend, and they are not shown in this case for the sake of simplicity.

### 5.2. Localization of the Neutral Axis

Based on the elastic beam theory, the neutral axis position corresponds to the reinforcements, pre-stresses strands, and concrete composite. It could be located by transforming the steel area in uncracked concrete into an equivalent concrete area [30]. Note that the concept of neutral axis in this paper refers to zero stress/strain change under applied load, which is not the absolute zero stress/strain location. The structural analysis shows the neutral axis in intact cross-section is located at the height of 100 cm from the bottom. When the cracks are fully developed in the tension zone and concrete completely lose its ability to resist tension forces, the neutral axis will move up to the height of 152.3 cm. As shown in Figure 11, the theory height of the neutral axis between the intact and fully cracked conditions range from 100 to 152.2 cm.

Then, for all available transducers at the upper (R1∼R4) and lower (R5∼R8) portions of the T-beam, lines connecting them are utilized to locate the coordinates of the stress invariant (*ε* = 0) through the linear intercepts of the *ε* diagrams. There are 4 × 4 receiver pairs for each of six source excitations, which implies 96 coordinates of the neutral axis locations can be determined at each loading step. Figure 12a and Figure 13a illustrate their location distributions and the mean heights under different loads where the localization results from gauge measurements and structural analysis are also denoted.

As illustrated in Figure 12, 10 strain gauges only estimate the neutral axis heights at two cross sections along the *x*-axis, whereas the localization results determined by 14 ultrasonic transducers almost cover the entire cross section. In Figure 12a and Figure 13a, there are two main differences among analysis and measurement results: (1) at the starting state (Figure 12a), the mean heights determined by gauge and wave measurements and are similar, but they are smaller than the analysis values. (2) The gauge-determined heights in Figure 12a and Figure 13a are always below the analysis results, and their mean heights slightly decrease as loads increase. However, the coda wave-determined heights vary around the analysis results along the *x*-axis, and their mean values increase linearly with loads. These phenomena are explained and discussed below.

In the region where the gauges are placed (i.e., the east face in Figure 5), there are no cracks existing and appearing during the experiments. However, the opposite west web contains several pre-existing micro-cracks, i.e., the whole structure is not completely healthy. According to linear theory and structural analysis, the neutral axis location moves up if the T-beam loses any part of its stiffness by cracking. The neutral axis locations determined by coda waves are obeying the laws, whereas the localization results from gauge measurements show the “move down” behavior, which is incorrect. This problematic phenomenon is also found in some other gauge observations in literature [4,6]. In multi-composite concrete, the issues of load redistribution, nonlinear thermal gradients, damages, corrosions, strain concentrations, and de-planation may contribute to non-uniform stretching behaviors at certain places. Since gauges only have detection capabilities at positions where they are exactly placed, their localization results only reflect the changes occurring in very limited areas of the undamaged surface. On the contrary, coda waves have traveled large volumes for a long transit time so that their neutral axis locations can indicate the global structural conditions not only limited to their installation positions.

### 5.3. Localization of Micro-Cracks

The mean heights of neutral axis determined by all available transducers evaluate global structural health. Further analysis with transducer networks, e.g., evaluate localization results from different source-receiver pairs, can locate inner micro-cracks up to a certain extent. Figure 13b shows an example of the mean heights under varied loads corresponding to each source excitation. The S1, S2, and S5 heights keep lower than the analysis value, whereas the S3 and S4 heights are always higher than the analysis value. All of their heights are varied in a limited range of 1∼3 cm as loads increase. Thus, the region between S3 and S4, either at the surface or inside concrete, contains damages. Additionally, the damages are not significantly worsened during the loading procedure. Furthermore, the S6 heights increase linearly as loads, and its height at 50 kN exceeded the associated S3∼S4 heights, which indicates that the outer part of the S6 position experiences the cracking effect during the loading procedure. These predictions are very consistent with our observations presented in Figure 5. In particular, the ultrasonic transducers are placed at the T-beam’s east face without cracks, but, at the opposite west web, one crack (crack 1) exists in the region between the S3∼S4 locations and two cracks (cracks 2 and 3 at the bottom) near S6 gradually get bigger under loading.

The localization results of neutral axis by a network of transducers indicate rough positions of multiple micro-cracks. More transducers and closer placement as well as analysis of different source-receiver pairs (i.e., sensor network) can lead to higher resolution. In a recent study, several other coda wave-based techniques such as maximum likelihood [23] and diffusion imaging [24,25,26] are developed for precise localization of cracks. Integrating them with this proposed technique will be an important task of our future work. Currently, an automatic detection system based on these techniques is being developed to simultaneously real-time monitor structural health and pinpoints crack positions.

## 6. Conclusions and Future Work

This paper describes an original technique to locate the neutral axis of large concrete structures by using coda wave measurements in a network of transducers. The applicability of the technique is demonstrated by monitoring a 30-m pre-stressed T-beam subjected to four-point bending tests causing stress changes. The experimental results show that this technique can locate the neutral axis at certain places without having to place transducer exactly there, and it is very efficient. Therefore, a neutral axis in a large region can be located in seconds with limited numbers of fixed transducers. The localization results indicate global structural health under varied loads as well as detect and locate multiple inner micro-cracks up to an acceptable extent.

In the real world, concrete structures are continuously subject to mechanical, chemical, and environmental effects that degrade their structural integrity. The neutral axis tied to material and geometry is directly correlated with the centroid of stiffness of a structure. This novel technique has the advantages of great efficiency, high sensitivity, easy operation, and low cost in neutral axis localizations. Implementation of it on in-service structures might promote novel NDT&E technology development for structural health monitoring and infrastructure asset management.

## Figures and Tables

**Figure 1 sensors-20-03895-f001:**
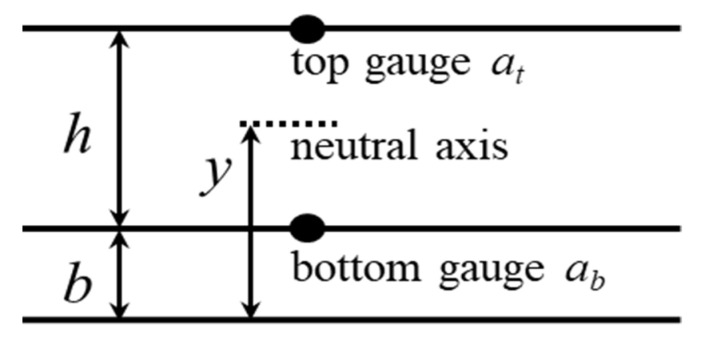
A cross-section with two parallel gauges measures strains at the top *a_t_* and at the bottom *a_b_*. By assuming linear strain distributions, the neutral axis height *y* is found as the intercept of the strain diagram *a_t_* and *a_b_*.

**Figure 2 sensors-20-03895-f002:**
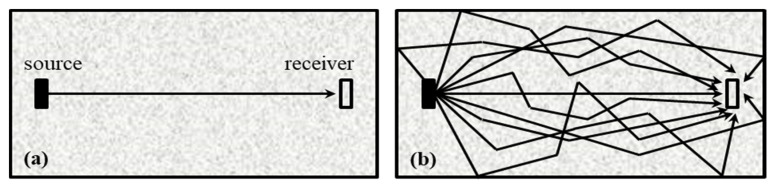
Principle of ultrasonic transmission measurements and propagation paths for (**a**) a direct wave and (**b**) a coda wave.

**Figure 3 sensors-20-03895-f003:**
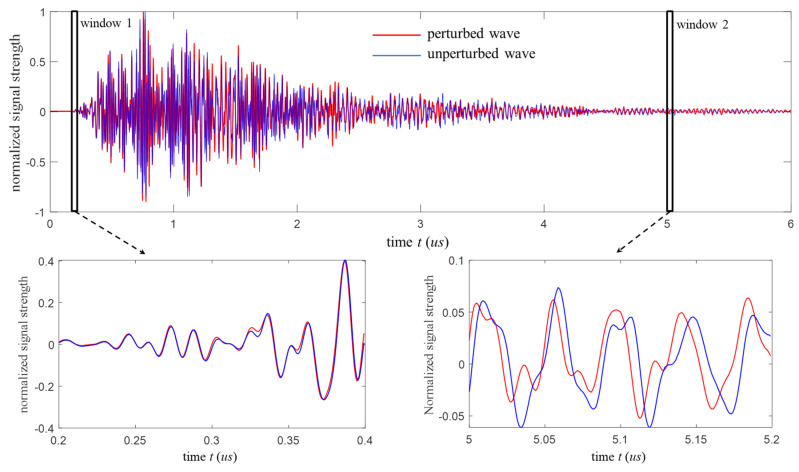
Ultrasonic wave measurements for the same geometry and sample without loads (unperturbed wave) and subjected to a small load (perturbed wave). While the direct waveforms are nearly invariable, coda waves at the late part of the signal show clear phase changes.

**Figure 4 sensors-20-03895-f004:**
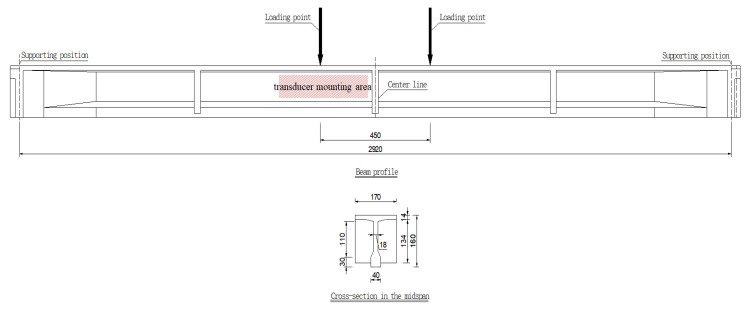
The front and right views of the concrete T-beam (unit: cm).

**Figure 5 sensors-20-03895-f005:**
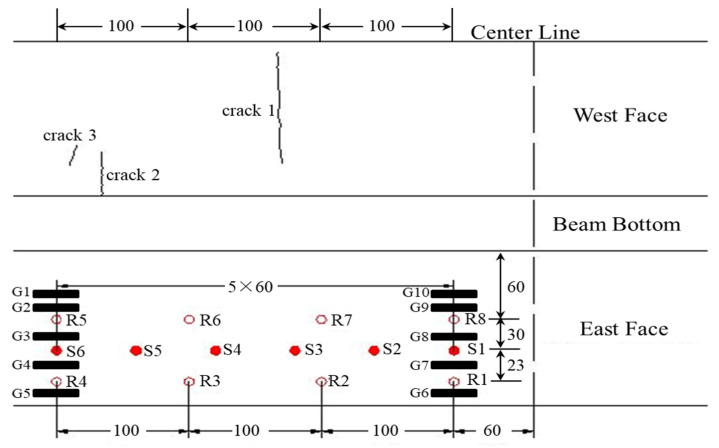
Schematic layout of the transducers and gauges networks (unit: *cm*).

**Figure 6 sensors-20-03895-f006:**
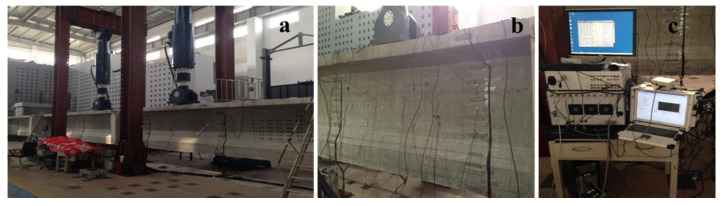
Photographs of (**a**) four bending test, (**b**) transducer and gauge networks, and (**c**) excitation and measurement equipment.

**Figure 7 sensors-20-03895-f007:**
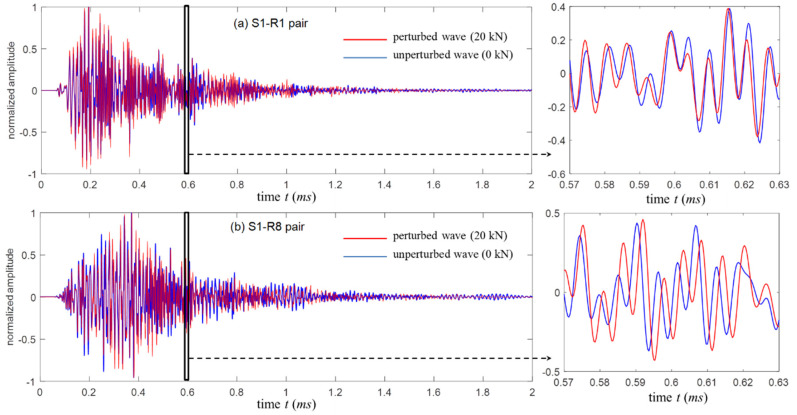
The averaged waveforms at 0 kN and 20 kN loads collected with: (**a**) the S1-R1 pair and (**b**) the S1-R8 pair.

**Figure 8 sensors-20-03895-f008:**
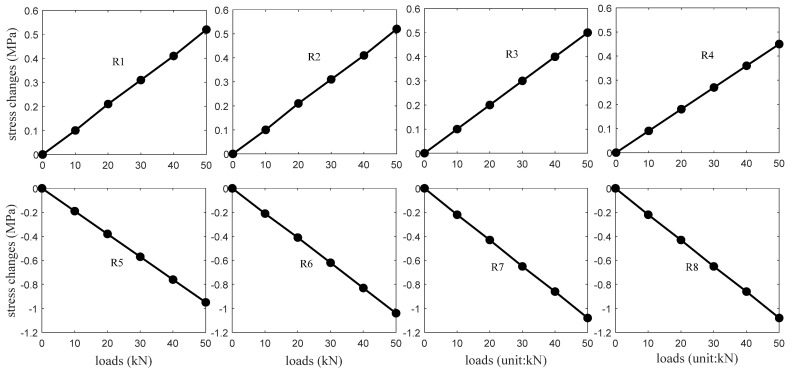
The structural analysis results of the stress changes at the R1∼R8 positions.

**Figure 9 sensors-20-03895-f009:**
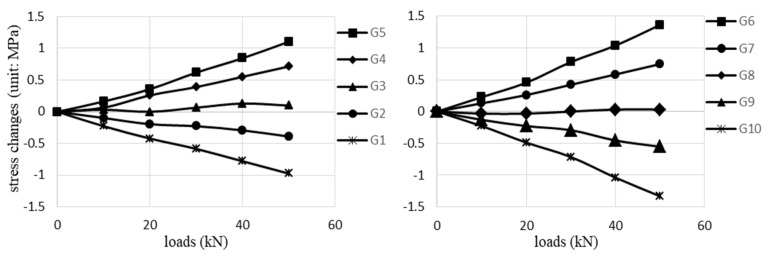
The values of stress changes measured by the strain gauge G1∼G10.

**Figure 10 sensors-20-03895-f010:**
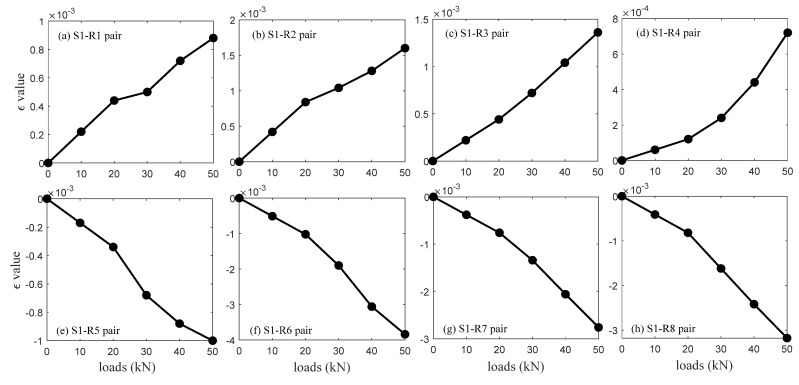
The *ε* values corresponding to the source S1 excitation and receivers R1∼R8 measurements.

**Figure 11 sensors-20-03895-f011:**
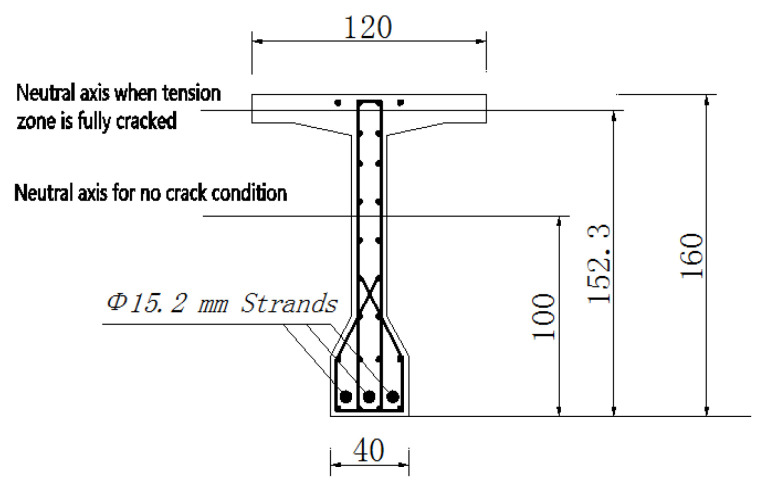
Reinforcements and pre-stressed strands in the cross section at the midspan (unit: cm).

**Figure 12 sensors-20-03895-f012:**
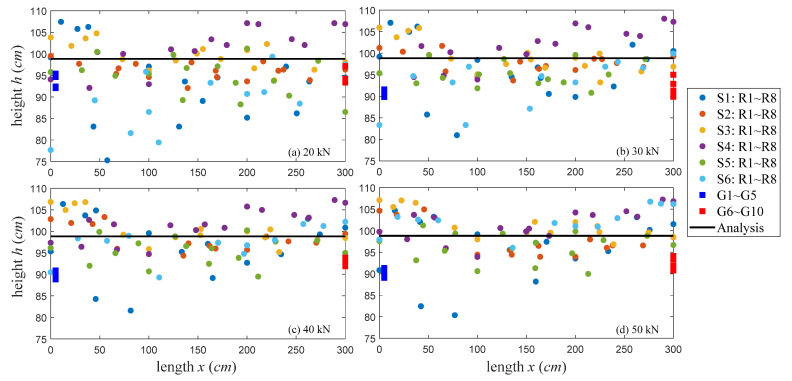
All the neutral axis locations determined by ultrasonic transducers, gauge gauges, and structural analysis. (**a**) The neutral axis locations at 20 kN; (**b**). The neutral axis locations at 30 kN; (**c**) the neutral axis locations at 40 kN, and (**d**). the neutral axis locations at 50 kN.

**Figure 13 sensors-20-03895-f013:**
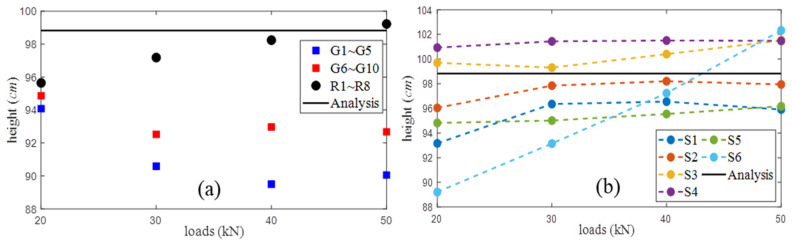
(**a**) The mean heights of neutral axis determined by transducers, gauges, and analysis, and (**b**) the neutral axis heights corresponding to each source excitation.

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
