# Peer review of "Condition Evaluation of an Existing T-Beam Bridge Based on Neutral Axis Variation Monitored with Ultrasonic Coda Waves in a Network of Sensors"

_sensors, 2020, doi:10.3390/s20143895_

Round 1

Reviewer 1 Report

an approach for structural health monitoring of concrete beam structures is presented using coda wave interferometry for detecting changes in the beam neutral axis.

The following minor corrections are required:

Typographical Comments
================
Lines 91, 205, 286: heading should begin in the next page

Technical Comments
=============
In the mathematical framework for the analysis of the coda wave presented in the paper, it is not clear how the neutral axis is determined from the normalised correlation coefficient between the unperturbed and perturbed coda wave.

Author Response

Dear Revier,

We would like to thank you for the comments and suggestions. Your critiques certainly helped to make this paper stronger.

Based on a number of related comments from all three reviewers, we reworked parts of sections 1 and 6 to present a literature review of coda wave-based techniques and further emphasize the novelty of our proposed approach. Several paragraphs in sections 2.1-2.2 are rewritten to clarify the algorithm for neutral axis localization. Figures 4 and 8 are deleted and Figures. 15-16 are combined to better show our experimental results. The revisions are highlighted with red color in the new work. We feel this new work is more useful to the researchers and the detailed responses corresponding to each comment can be found in the attached file.

Author Response

(The authors gave the same response as above.)

Reviewer 3 Report

The paper shows a very interesting way of SHM in concrete Beams. I did not find any mayor issues in the paper.

  • The description of the x axis in most plots changes sometimes. I guess (unit:s) for example is not intended but rather (s) as done in figure 16.

  • Figure 6: please add positions to the upper picture.

Furthermore i would be highly interested to study the influence of outer condition changes on the coda waves in future research. I can think of a negative influence of for example water on one side on the sensitivity for this technique.

Author Response

(The authors gave the same response as above.)

Round 2

Reviewer 2 Report

I can now recommend the paper for publication.